# Excretion of Ni, Pb, Cu, As, and Hg in Sweat under Two Sweating Conditions

**DOI:** 10.3390/ijerph19074323

**Published:** 2022-04-04

**Authors:** Wen-Hui Kuan, Yi-Lang Chen, Chao-Lin Liu

**Affiliations:** 1Department of Safety, Health and Environmental Engineering, Ming Chi University of Technology, New Taipei 24301, Taiwan; whkuan@mail.mcut.edu.tw; 2Chronic Diseases and Health Promotion Research Center, Chang Gung University of Science and Technology, Chiayi 61363, Taiwan; 3Department of Industrial Engineering and Management, Ming Chi University of Technology, New Taipei 24301, Taiwan; 4Department of Chemical Engineering, Ming Chi University of Technology, New Taipei 24301, Taiwan; clliu@mail.mcut.edu.tw; 5Department of Chemical and Materials Engineering, Chang Gung University, Taoyuan 33302, Taiwan

**Keywords:** sweating condition, heavy metal excretion, dynamic exercise, hot environment

## Abstract

Physiologists have long regarded sweating as an effective and safe means of detoxification, and heavy metals are excreted through sweat to reduce the levels of such metals in the body. However, the body can sweat through many means. To elucidate the difference in the excretion of heavy metals among sweating methods, 12 healthy young university students were recruited as participants (6 men and 6 women). Sweat samples were collected from the participants while they were either running on a treadmill or sitting in a sauna cabinet. After they experienced continuous sweating for 20 min, a minimum of 7 mL of sweat was collected from each participant, and the concentrations of nickel (Ni), lead (Pb), copper (Cu), arsenic (As), and mercury (Hg) were analyzed. The results demonstrated that the sweating method affected the excretion of heavy metals in sweat, with the concentrations of Ni, Pb, Cu, and As being significantly higher during dynamic exercise than during sitting in the sauna (all *p* < 0.05). However, the concentrations of Hg were unaffected by the sweating method. This study suggests that the removal of heavy metals from the body through dynamic exercise may be more effective than removal through static exposure to a hot environment.

## 1. Introduction

Research on sweating has mostly focused on the differences in the sweat composition among different groups and environments. When examining the effects of sex and maturity stage (prepuberty, adolescence, and young adulthood) on electrolyte (chloride, sodium, and potassium ions) loss in participants exercising in a hot environment, the percentage of electrolyte loss in young adults was higher than that in the other two groups with respect to unit weight, whereas no significant difference in the sweat composition was found between the male and female groups [1]. Another study also found no difference in iron and zinc (Zn) loss between the sexes through sweat during prolonged exercise of 2 h [2]. Higher concentrations of magnesium and calcium ions may exist in the sweat of individuals exercising in hot environments than in those exercising in indoor and outdoor environments, although chloride, sodium, and potassium ion concentrations were identical among the sweating conditions [3].

In the human body, heavy metals, such as mercury (Hg), cadmium (Ca), lead (Pb), chromium (Cr), and metalloid-like arsenic (As), generally exhibit biological toxicity. These heavy metals cannot be decomposed in water, and when they are consumed through contaminated water, their toxicity is amplified, and the metals combine with other toxins in the water to form more toxic organic substances. Studies have discovered Pb in children’s toys [4], As in rice [5], aluminum in kitchen pots [6], and Hg in fish [7]. This indicates that different aspects of people’s daily lives, including air, soil, water, and food, are conduits for bioaccumulated toxic elements. Many heavy metals are more easily excreted through sweat; sweating is thus proposed as a preferred method for detoxification [8]. Sears et al. [9] surveyed 122 studies and analyzed 24 of them to study heavy metals in sweat. They reported that the study samples, methods of sweat collection, and heavy metal concentrations were all divergent among studies. However, sweat remains an effective means of evaluating heavy metal concentrations because some heavy metals were detected in higher concentrations in sweat than in blood and urine; for instance, Ni, Pb, and Cr in sweat have been reported to be 10 to 30 times than that in blood and urine [8].

Although sweating has been recognized by the medical community as an effective and safe method for detoxification, the conditions under which individuals sweat are varied and may lead to differences in heavy metal excretion. It was suggested that sauna therapy can enable the elimination of heavy metals from the body [10]. Tang et al. [11] found that five heavy metals (Ca, Pb, Cr, Zn, and copper (Cu)) in sweat exhibited higher levels than those in urine after arduous exercise and indicated that sweating through exercising is able to remove detrimental heavy metals effectively from the human body. They also identified exercise conditions as the most favorable for inducing sweating. Furthermore, higher excretions of electrolytes and urea were observed during dynamic running than during sitting in a static thermal environment [12].

Previous findings indicate that an effective means of inducing sweating to optimize the amount of excreted heavy metals should be identified; however, studies have rarely systematically compared the influence of the different stimuli of sweating on the concentration of heavy metals in sweat. The objective of this study was thus to preliminarily examine the difference in the excretion of heavy metals under two sweating conditions. This study performed a simulated sweating test on 12 healthy young participants (6 men and 6 women) and collected their sweat under two sweating conditions (dynamic running and a static sauna environment). Sweat compositions were analyzed for the excretion of different heavy metals (Ni, Pb, Cu, As, and Hg) under the two conditions. We hypothesized that the excretions of five heavy metals in sweat under two sweating environments could be distinct, based on different physiological mechanisms.

## 2. Materials and Methods

### 2.1. Participants

Twelve young university students (6 men and 6 women) were recruited as participants; the mean (standard deviation) age, height, and body mass index of the male participants were 21.8 (0.4) years, 169.2 (1.3) cm, and 59.6 (7.3) kg, respectively, and the corresponding data for the female participants were 22.0 (0.5) years, 160.6 (5.1) cm, and 53.2 (5.4) kg, respectively, as listed in Table 1. All participants reported that they were moderately physically active, were healthy and asymptomatic of illness, and had no pre-existing injuries. To enable the general application of the results, the participants’ maximum oxygen uptake was set at approximately fair to good levels [13]. All participants dined in the same restaurant in the year preceding the experiment; for 1 week prior to the experiment and over its course, the participants’ food intake was controlled to ensure that they all ate the same food. This minimized any biases resulting from the variation in daily individual food intake and minimized the influence of the homogeneity within the group. Furthermore, the participants were not allowed to take medication over the course of the experiment. Those who did so would be excluded. The experimental procedures were approved by the Ethics Committee of National Taiwan University, Taiwan (ethical code: 202012EM025), and all study participants provided written consent prior to the start of the experiment.

### 2.2. Sweating Inducement and Environmental Conditions

The testing setup and sweat-collection procedure were primarily developed with reference to our previous study [12]. Two sweating-inducing conditions were implemented by the study, including running on a treadmill (CS-5728, Chanson, Taipei, Taiwan) and inactive overheating in a heat sauna cabinet. The temperatures were set at 25 °C and 45 °C for the running room and sauna cabinet, respectively, whereas humidity was set at a 40% level for both conditions. The controllable range and incremental interval of temperature in the utilized sauna cabinet were 40–60 °C and 5 °C, respectively. The identical humidity levels were set up to prevent differences in sweat evaporation levels between the two sweating conditions. To examine the influence of the sweat rate on the sweat composition generated by the two conditions, the sweat rate was measured and calculated through a pretest. The sweat rates under the two sweating conditions were determined using the weighing method; they were nearly equivalent at 1.8 mg/cm^2^/min during the 20-min running (or overheating) period and the 30-min collection extension period to ensure a sufficient collected sweat volume (>7 mL) for further analysis.

During running, the participants were requested to run on a treadmill; the pace was increasingly speeded from 5 to 10 km/h during the first 10 min, followed by an additional 10 min of running. In the inactive overheating condition, the sweat test was performed in a sauna cabinet for 20 min. The participants’ sweat was collected until 7 mL of sweat had been successfully accumulated from their upper backs.

### 2.3. Sweat Collection

Studies have adopted varying methods to collect sweat for composition analyses [14,15,16], including the direct collection of sweat from the skin using glass jars or tubes and sweat collection using made-to-measure apparatuses (e.g., sweat pouches, glass pipettes, and arm bags) and commercial products. However, pH alteration, skin irritation, barrier property disturbance, and variations in the body region for sweating and the aim of examination have resulted in difficulty in designing a universal apparatus to fit all test conditions [17].

This study employed a laboratory-made sweat collector comprised of a funnel with a glass tube, which was cleaned using deionized water, air-dried, and then covered with tin foil to prevent contamination. Collectors were prepared and coded before the tests based on the required sample volume. The sweat samples of the participants were obtained using the sweat collector capable of collecting sufficient volumes of sweat (i.e., 7 mL) with minimal skin interference for the required duration, as in our previous study [12]. During sweat collecting, an experimenter used the funnel edge to separately scrape the sweat from the upper backs of participants as conducted by the previous study [18].

### 2.4. Sweat Preservation and Composition Analysis

In the test, sweat samples were promptly filtered using a syringe filter (0.22-μm pores) to remove cellular debris for each trial, to avoid small particles in liquid samples becoming trapped during pulverization for inductively coupled plasma-atomic emission spectroscopy (ICP-AES) and inductively coupled plasma-mass spectrometry (ICP-MS). The 0.22-μm pores were smaller than the size of skin flakes (>10 μm), as suggested by Mackintosh et al. [19]. The filtrate was collected in a capped glass tube and stored at 4 °C. All composition analyses were acidified with HNO_3_ and completed within 32 h of each trial. The sweat samples were analyzed for Ni, Pb, Cu, As, and Hg levels, complying with the standard methods [20,21]. Samples were measured in duplicate. Ni, Pb, and Cu concentrations were analyzed using ICP-AES (PerkinElmer, Avio 200, Waltham, MA, USA) with method detection limits (MDLs) of 1.1, 3.0, and 1.5 μg/L, respectively. Analyses of As and Hg were performed using ICP-MS (Agilent 7800, Santa Clara, CA, USA) with MDLs of 0.03 and 0.05 μg/L, respectively. Method detection limits (MDLs) for each metal were conducted in our laboratory following the standard methods [22,23]. MDLs were calculated as three times the standard deviation of a set of method blanks and the values of quantification limits of the study. The calibrated standard solutions of Ni, Pb, and Cu were prepared, ranging from 0–500 μg/L; that of As and Hg from 0–5 μg/L, which ranges covered the concentration of metals in sweat found in the literature [15,24].

### 2.5. Experimental Design and Procedure

During the experiment, the sweat produced under two testing conditions was collected from the target sample regions (8 cm × 8 cm) using the laboratory-made sweat collector within the 20-min running (or overheating) and an extended sampling period (i.e., 30-min). Subsequently, a sufficient sweat sample (i.e., 7 mL) was successfully collected from the upper back of each participant for every normal 50-min duration.

Before the test, participants were requested to control their exercise and diet for 1 week. Two hours before the experiment, the participants were asked to urinate and fast and were only allowed to intake 200 mL of water. Before testing began, the experimenter cleaned the participants’ upper backs using a brush and tap water to remove any dirt and skin residue. Deionized water was then used to wash the skin region a second time to ensure the skin was clean. When using an arm-bag technique, the skin surface may become contaminated by skin desquamation and trace mineral residues from dirt and other minerals under the fingernails that are difficult to remove, even with meticulous cleaning [25]. Because this cleaning procedure was performed before running the test, the trace mineral concentrations of the sweat sample on the upper back remained unchanged during 3 h of exercise. Therefore, in our test, a demanding cleaning procedure could be conducted to minimize possible skin contamination because the sweat sampling areas were easy to clean.

To prevent cumulative fatigue, the tests were separated by a two-day interval and were performed in a counter-balanced arrangement. During testing, the 12 participants were randomly divided into 2 groups (groups A and B), comprising 6 participants each. On the first day, the participants in groups A and B were asked to perform dynamic running and sauna cabinet sitting trials, respectively, for collecting sweat samples of 7 mL (Figure 1). On the second day, groups A and B performed sauna cabinet sitting and dynamic running trials, respectively. A total of 24 sweat samples were collected (12 participants × 2 sweating conditions).

### 2.6. Statistical Analysis

The sweat compositions of the concentrations of Ni, Pb, Cu, As, and Hg were measured under two sweating conditions (running vs. inactive overheating). A paired *t*-test was used to investigate the effects of environmental variables (treadmill and sauna cabinet) on the sweat composition to prevent the varying levels of heavy metals among individuals from interfering with the accuracy of the test. Statistical analyses were performed using SPSS Statistics 23.0 (IBM, Armonk, NY, USA). An alpha level of 0.05 was used to indicate the minimum level of significance.

## 3. Results

The results of a paired *t*-test of the sweat compositions under two sweating conditions are presented in Table 2. As indicated in the table, Ni, Pb, Cu, and As concentrations were higher after the participants ran on the treadmill than after the participants sat in the sauna cabinet. This indicates that dynamic exercise may lead to the excretion of a larger amount of heavy metals, whereas a static hot environment may cause a higher proportion of water in sweat. However, Hg did not exhibit a significant difference between the two sweating conditions. Figure 2 further illustrates the boxplots of sweat compositions for five heavy metals under two sweating conditions. Because the measurement amounts among heavy metals are quite different, two scales are thus adopted in the figure. It can be seen that there are significant differences in sweat excretions between the dynamic running and static heat conditions for four of the five heavy metals. In addition, there were relatively large individual differences that existed in heavy metal excretions in the sweat of the participants.

## 4. Discussion

The results demonstrated that the sweating method influenced the excretion of heavy metals in sweat and, as hypothesized, the concentrations of Ni, Pb, Cu, and As were significantly higher during dynamic running than during sitting in the sauna. After strenuous exercise, sweat has generally a higher capacity than urine to remove heavy metals from the body. Tang et al. demonstrated that dynamic exercise can not only considerably affect the balance of trace elements but can also effectively remove toxic heavy metals from the body and reduce the accumulation of trace heavy metals [11]. However, that study only investigated the sweating condition of dynamic exercise. We employed previously, the same two conditions as those used in this study to investigate the excretion of urea, uric acid, and electrolytes in collected sweat samples; and discovered that the secretions of urea and K^+^ were significantly higher during running than during inactive overheating [12]. Although the concerned sweat composition in this study was different from our previous study, the current findings revealed significant differences in the concentrations of heavy metals in sweat between dynamic exercise and static thermal conditions.

Heavy metals are most abundant exogenously, but small amounts enter the body through food and water. Most are barely metabolized, and their major expellant routes are urine and sweat. Some heavy metals, however, such as iron, Cu, and Zn, are important for life and metabolism. Others, however, are toxic, interfering with protein function and enzyme activities. Excessive accumulation of toxic heavy metals can even lead to chronic poisoning or death [26]. Knowledge of the mechanisms underlying the accumulation and excretion of toxic elements in the human body remains limited [11]. Heavy metals may be excreted in appreciable quantities through the skin, and the excretion rates through the skin could be comparable with or even exceed those of urinary excretion in a 24-h period [9]. Although the recommended sweating methods for detoxification usually include both exercise [10,11] and heat [27,28], to the best of our knowledge, no comparative study of the two sweating methods has been conducted.

The concentrations of various heavy metals in sweat obtained from the treadmill group and sauna group likely differed because each metal is excreted through its own physiological mechanism and dilution factors. Although the core temperature of homeotherms fluctuates with diurnal rhythms, the variation is generally less than 2 °C [29]. In humans, the thermoregulatory center is located in the hypothalamus and maintains the core temperature to approximately 37 °C through conduction, convection, radiation, and evaporation of sweat [30,31]. This core temperature generally increases with intense activity such as exercise [32,33,34]. However, the core temperature is only slightly affected by environmental temperatures. Although the core temperature has been reported to reach 40 °C with long-distance running, it increased by only 0.75 °C when a normal subject remained in a room at a temperature of 46 °C for 4 h [35,36,37]. During exercise, the heart rates increase, and the elevation of the core temperature triggers faster circulation. The warmer blood from the body’s core is directed to the skin through blood perfusion, and the heat is released with sweat. The contents of sweat are expelled from the blood directly through sweat secretion [32,36]. However, when the ambient temperature rises above the core temperature, the core temperature increases, and only local heat dissipation accompanied by behavior patterns for heat removal, such as rolling up sleeves or applying an ice pack, occurs. Generally, local cutaneous vessels dilate to increase blood flow and body fluid penetration, and hypotonic sweat with a few metals is secreted for heat dissipation [36]. This explains why the concentrations of heavy metals excreted through sweat due to hyperthermic surroundings (sauna group) were lower in this study.

The heavy metals in sweat are highly correlated to those in the serum. The concentrations of heavy metals in serum vary with the region because of dietary exposure and because metal ions may be bound to proteins or red blood cells (RBC). Ni can be bound to albumin, α-2 macroglobulin, or histidine [38], and 90% of Hg in the human body is stored within RBC as methylmercury [39,40,41]. The selectivity and mechanism of such excretion remain unclear. Among the five metal ions in this study, Pb and Cu levels are generally much higher in Taiwan than in Western countries [42]. This is likely because the distribution of both metals in Taiwanese rivers is the highest in the world [43] and because contaminated seafood is somewhat common in Taiwan. However, Hg concentrations in sweat are relatively low compared with the concentrations of other heavy metals because only 10% of the total concentration of Hg in the body is inorganic, and half of this inorganic Hg is distributed in plasma [39,40,41]. Therefore, only trace Hg ions are secreted from the blood and are detectable in sweat.

Variations in measured concentrations were likely the result of differences in excretion among individuals. Generally, heavy metal intake in humans originates from two major sources. The first source is the mother during infancy, and the second is daily exposure [11,42,43]. These variable sources may lead to considerable individual differences in heavy metal compositions. In addition, the experimental setting also determined the interpretative data. Tang et al. [11] found the excretions of Cu 2.59 μg/L and Pb 0.63 μg/L in sweat under arduous exercise, while Genuis et al. [24] obtained those of Cu 652.49 μg/L and Pb 25.67 μg/L under infrared or steam sauna. The large variance in excretion of each metal between the studies demonstrated that not only the individual differences but also the systematic experiments influenced the results. Therefore, in this study, the experimental design was simplified as much as possible to prevent the interference of the aforementioned factors while enabling the identification of potential variables. Twelve physically healthy male and female participants were recruited, and comparisons of the excretion of heavy metals in sweat samples were made at the individual level through the paired *t*-test to identify differences between the concentrations under the two sweating conditions and to eliminate the influence of individual differences [9].

Although sex differences were not explored in this study due to the relatively small sample size, the results revealed that Ni excretion was higher in women than in men. This may be because the Ni concentrations in the body of these female participants (mean ± standard deviation: 81.0 ± 35.2 μm/L) were originally higher than those of the male participants (33.5 ± 54.3 μm/L). The high standard deviations indicated a high variability among participants and may be attributed to the relatively small sample size. In addition, this may have been affected by Ni concentrations in some sweat samples of the sauna group being lower than the respective MDL. A similar phenomenon occurred with Cu excretion. Differences in heavy metal concentrations between the sexes, both within the body or those excreted, warrant further study. Because this study focused on the comparison of heavy metal concentrations between two sweating environments, the interference of other variables was minimized.

Because the number of studies that have analyzed the concentrations of trace heavy metals in the body through sweat analysis is limited, further research in this area is required. This preliminary study suggests that individuals experiencing adverse effects due to toxic elements or those who have regular exposure to or accretion of toxicants may consider sweating through dynamic exercise (e.g., running) as an option. However, only 12 healthy young university students were recruited; this sample size is relatively small. A more extensive study would allow for an increase in experimental settings (various populations, health statuses, and sweating conditions) for future research.

## 5. Conclusions

Preliminary studies reported here found that the concentrations of heavy metals (Ni, Pb, Cu, and As) in sweat after dynamic exercise were higher than those in sweat after experiencing a thermal sauna environment, indicating that different sweating methods may affect the excretion of heavy metals in sweat. The results can serve as a reference for sweat detoxification while performing daily activities.

## Figures and Tables

**Figure 1 ijerph-19-04323-f001:**
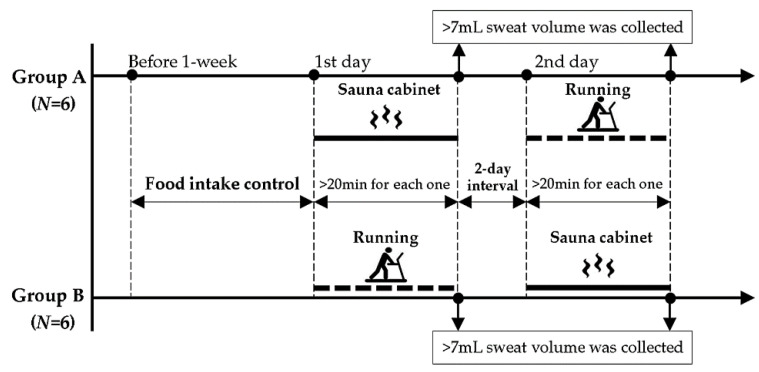
Schematic timeline of study design.

**Figure 2 ijerph-19-04323-f002:**
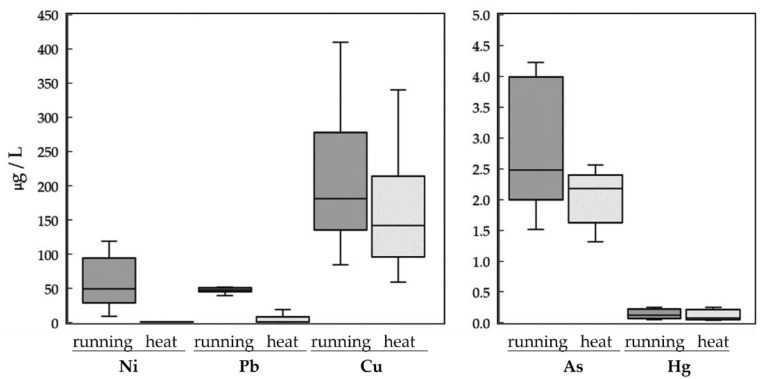
Boxplots of sweat excretions for 5 heavy metals under two sweating conditions.

**Table 1 ijerph-19-04323-t001:** Age, physical characteristics, and exercise habits of the participants.

	Men (*N* = 6)			Women (*N* = 6)		
Items	Mean SD Range			Mean SD Range		
Age (years)	21.8	0.4	21.2–22.0	22.0	0.5	21.0–23.3
Stature (cm)	169.2	1.3	168.0–172.5	160.6	5.1	153.0–171.8
Body mass (kg)	59.6	7.3	50.0–68.4	53.2	5.4	45.5–60.0
Resting heart rate (beats/min)	74.2	3.2	68–79	72.3	3.0	68–77
Maximum O_2_ uptake (mL/kg/min)	43.0	1.6	41.0–45.0	34.2	1.3	33.0–36.0
Exercise habit (times/week)	3.2	0.8	2–4	1.4	0.5	1–2

Notes: SD—standard deviation.

**Table 2 ijerph-19-04323-t002:** Sweat compositions were investigated using the paired *t*-test under two sweating methods (*N* = 12, unit in μg/L).

	Ni	Pb	Cu	As	Hg
Treadmill	57.3 (36.4)	52.8 (15.9)	206.5 (99.8)	2.9 (1.0)	0.3 (0.4)
Sauna cabinet	5.2 (14.1)	4.9 (6.6)	159.4 (88.5)	2.1 (0.4)	0.2 (0.4)
Difference	52.1 (34.8)	47.9 (17.8)	47.1 (76.9)	0.8 (0.9)	0.1 (0.2)
*t* value	4.623	9.638	3.198	2.900	0.960
Significance	*p* < 0.001	*p* < 0.001	*p* < 0.01	*p* < 0.05	*p* = 0.358

Notes: Data in mean (standard deviation).

## Data Availability

The data are available upon reasonable request to the corresponding author.

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
