# Peer review of "Excretion of Ni, Pb, Cu, As, and Hg in Sweat under Two Sweating Conditions"

_ijerph, 2022, doi:10.3390/ijerph19074323_

Round 1

Reviewer 1 Report

This article reports the results of a single study of 6 young mean and 6 young women, during which 7 ml of sweat were collected under two kinds of conditions that induce sweating: rigorous treadmill exercising versus inactive, in a sauna; and the sweat was analyzed for some beneficial and toxic heavy metals.  Much larger levels were found with the exercise sweat in the case of Ni and Pb, with smaller differences (higher in exercise sweat) for Cu and As, and no differences in the case of Hg.  Side-by-side comparisons of metal concentrations from two kinds of sweating, in the same subjects, has not been done previously.  The methods used for these studies, their calculations and data seem solid.  Generally, the descriptions and narrative are useful, though a bit overdone and repetitive.  Some clarification of statements and methods is also needed.

It is recommended that the authors go over the discussion of their data in the Results and Discussion section itself, and in relation to what is described in the Introduction, to eliminate repetition, and generally make the narrative more succinct.

Specific additional points are:

Line 64-65:  this statement (higher metal concentrations higher in sweat than blood) is debatable and must be followed up with references to actual data (or eliminated).  For example, for Cu, this is not at all true.

Lines 113 vs. lines 114 and 115: Please explain whether the subjects were exposed to 45 degrees, or to increasing temperatures between 40 and 60 degrees.

Lines 177-118:  Explain what was referred to as a pretest – was that run on all subjects, or what?   What was important about it?

Line 136: “comprised OF a funnel”

Line 143: “which was determined with reference” – do you mean it was done based on what was reported in the reference?

Line 146 “sweat samples…promptLY filtered”

Line 168: “the skin WAS clean”   “using AN arm bag”

Line 171: “performed BEFORE RUNNING THE test”

Lines 211-213: the two studies did not both look at urea and K+ (please correct)

Lines 215-216: This sentence is not correct.  Biologically, heavy metals are not unavailable?? And metals of course are not degradable (being atoms).  They do not generally “destroy” protein and enzyme activity – Pb does inhibit one enzyme activity, but does a lot of other things to proteins, for example; Cu and Ni do not do such things in general.

Line 220: “the excretion Rates were considered to march….”  What is actually the truth here?

Line 227: “provided prelimiNARY findings”

LINE 228: It is suggested to state “significantly changed the excretion”

Line 254: Instead of “consequently”, suggest saying “This explains why…”

Line 256:  What components of sweat are similar to those of blood?  And are you referring to whole blood or plasma?

Line 306: Suggest this sentence begin with “Preliminary studies reported here found…”

Author Response

Response to Reviewer 1 Comments
The authors wish to sincerely appreciate the reviewers for the opportunity given to
improve our manuscript. Please find below the improvement and explanation made
by the authors based on the reviewer’s comments and recommendations.
※ The marked changes were marked by using colored (red) text in the revised manuscript.
This article reports the results of a single study of 6 young men and 6 young women,
during which 7 ml of sweat were collected under two kinds of conditions that induce
sweating: rigorous treadmill exercising versus inactive, in a sauna; and the sweat was
analyzed for some beneficial and toxic heavy metals. Much larger levels were found
with the exercise sweat in the case of Ni and Pb, with smaller differences (higher in
exercise sweat) for Cu and As, and no differences in the case of Hg. Side‐by‐side
comparisons of metal concentrations from two kinds of sweating, in the same subjects,
has not been done previously. The methods used for these studies, their calculations
and data seem solid. Generally, the descriptions and narrative are useful, though a
bit overdone and repetitive. Some clarification of statements and methods is also
needed.

Point 1: It is recommended that the authors go over the discussion of their data in the
Results and Discussion section itself, and in relation to what is described in the
Introduction, to eliminate repetition, and generally make the narrative more succinct.
Response 1: Thanks for the valuable comment. In the revision, we have separated the
original Results and discussion into Results section and Discussion section
independently. The Discussion section has been also reorganized and rewritten to the
more succinct narrative based on the results what we really obtained in the study.
Specific additional points are:

Point 2: Line 64‐65: this statement (higher metal concentrations higher in sweat than
blood) is debatable and must be followed up with references to actual data (or
eliminated). For example, for Cu, this is not at all true.
Response 2: Following the reviewer’s suggestion, the sentence has been rephrased in
the revision and actual data indicated in reference were cited as follows.
However, sweat remains an effective means of evaluating heavy metal concentrations
because some heavy metals were detected in higher concentrations in sweat than in
blood and urine; for instance, Ni, Pb, and Cr in sweat have been reported 10 to 30
times than that in blood and urine [8]. (Line 52‐55)

Point 3: Lines 113 vs. lines 114 and 115: Please explain whether the subjects were
exposed to 45 degrees, or to increasing temperatures between 40 and 60 degrees.
Response 3: Subjects were exposed to 45°C during the studied scenario of heatstimulated
sweating. The controllable range and incremental interval of temperature
in the utilized sauna cabinet are 40‐60°C and 5°C, respectively. The sentence has been
revised to clarify the temperature of respective conditions as suggested. (Line 107‐108)

Point 4: Lines 177‐118: Explain what was referred to as a pretest – was that run on all
subjects, or what? What was important about it?
Response 4: In order to smoothly conduct the sweat collection and obtain enough
volume for composition analysis and, a pretest had been performed before the study.
The pretest did not run on all but several subjects participating the study and
experimenters who collected sweat, recorded and thus analysed the data. The pretest
helps the experimenters and participants familiar with the procedures and verify
sweat amount for the subsequent analyses. (Line 110‐111)

Point 5: Line 136: “comprised OF a funnel”
Response 5: Revised, as suggested. (Line 129)

Point 6: Line 143: “which was determined with reference” – do you mean it was done
based on what was reported in the reference?
Response 6: This unclear sentence has been revised in the revision. During sweat
collecting, an experimenter used the funnel edge to separately scrape the sweat from
the upper backs of participants as conducted by the previous study [18]. What we
referred from the study were sweat collection and body region that was collected.
(Line 135‐136)

Point 7: Line 146 “sweat samples…promptLY filtered”
Response 7: Revised, as suggested. (Line 138)

Point 8: Line 168: “the skin WAS clean” “using AN arm bag”
Response 8: Revised, as suggested. (Line 167)

Point 9: Line 171: “performed BEFORE RUNNING THE test”
Response 9: Revised, as suggested. (Line 170)

Point 10: Lines 211‐213: the two studies did not both look at urea and K+ (please
correct)
Response 10: Sorry for the unclear description. This part has been rewritten to a more
clear description as follows.
Although the concerned sweat composition in this study was different from our
previous study, the current findings revealed significant differences in the
concentrations of heavy metals in sweat between dynamic exercise and static thermal
conditions. (Line 226‐228)

Point 11: Lines 215‐216: This sentence is not correct. Biologically, heavy metals are
not unavailable?? And metals of course are not degradable (being atoms). They do
not generally “destroy” protein and enzyme activity – Pb does inhibit one enzyme
activity, but does a lot of other things to proteins, for example; Cu and Ni do not do
such things in general.
Response 11: The comment is really appreciated. The sentence has been rephrased as
below to express clearly.
Biologically, heavy metals are exogenous and mainly uptaken with water. They are
barely metabolized and their major expellant routes are urine and sweat. Some heavy
metals, such as iron, Cu, and Zn, are involved in biological chemistry reaction
physiologically. However, some are toxic on interfering protein functions and enzyme
activities. Excessive accumulation of toxic heavy metals can potentially lead death or
to chronic poisoning [26]. (Line 229‐234)

Point 12: Line 220: “the excretion Rates were considered to march….” What is actually
the truth here?
Response 12: Thanks for your advice. We have revised the sentence as follows.
Heavy metals may be excreted in appreciable quantities through the skin, and the
excretion rates through the skin could be comparable with or even exceed those of
urinary excretion in a 24‐h period [9]. (Line 235‐237)

Point 13: Line 227: “provided prelimiNARY findings”
Response 13: As suggested by the reviewers, the Discussion should be made the
narrative more succinct, this part has been removed from the manuscript.

Point 14: LINE 228: It is suggested to state “significantly changed the excretion”
Response 14: As suggested by the reviewers, the Discussion should be made the
narrative more succinct, this part has been removed from the manuscript.

Point 15: Line 254: Instead of “consequently”, suggest saying “This explains why…”
Response 15: Revised, as suggested. (Line 260)

Point 16: Line 256: What components of sweat are similar to those of blood? And are
you referring to whole blood or plasma?
Response 16: We are very grateful for the comment and sorry for the confusion. The
sentence has been revised as below.
The heavy metals in sweat are highly correlated to those in the serum. (Line 262)

Point 17: Line 306: Suggest this sentence begin with “Preliminary studies reported
here found…”
Response 17: Revised, as suggested. (Line 312)

Reviewer 2 Report

Review comments on ijerph-1631176: Excretion of Ni, Pb, Cu, As, and Hg in Sweat under Two Sweating Environments

The manuscript described the analyses of different heavy metals in the sweat of 12 healthy participants under two sweating conditions (dynamic running and a static sauna environment). The authors found that Ni, Pb, Cu, and As excreted in sweat under dynamic running were higher than those under static sauna. Overall, the study was interesting and appropriately designed. However, this was just a preliminary study to investigate the effects of sweating conditions on the secretion of heavy metals since the sample size was small (as a result, the SDs were high). Although the authors admitted this limitation in the manuscript, there are other issues, as follows.

  1. There were previous studies reporting similar data (as mentioned in the Introduction section). The authors should highlight the novelty of this study and its contribution to the field.
  2. The results should be compared with those in previous studies where relevant.
  3. Analysis methods for each metal should be in detail (lines 152-156). Relevant references should be cited.
  4. Please clarify the MDLs (lines 154-156). Were they from previously reported data? Were there values of quantification limits?
  5. Table 2: please clarify the numbers presented in a table footnote. Were they Means (SDs)?
  6. The data in Figure 2 should be presented as boxplots, which clearly show the range, median, and mean values of each dataset. It will be more informative to readers.
  7. The data for Hg were unreliable. The samples should be preconcentrated before analysis, by which those concentrations below MDLs could be measured accurately.

Author Response

Response to Reviewer 2 Comments
The authors wish to sincerely appreciate the reviewers for the opportunity given to
improve our manuscript. Please find below the improvement and explanation made
by the authors based on the reviewer’s comments and recommendations.
※ The marked changes were marked by using colored (red) text in the revised manuscript.
The manuscript described the analyses of different heavy metals in the sweat of 12
healthy participants under two sweating conditions (dynamic running and a static
sauna environment). The authors found that Ni, Pb, Cu, and As excreted in sweat
under dynamic running were higher than those under static sauna. Overall, the study
was interesting and appropriately designed. However, this was just a preliminary
study to investigate the effects of sweating conditions on the secretion of heavy metals
since the sample size was small (as a result, the SDs were high). Although the authors
admitted this limitation in the manuscript, there are other issues, as follows.
Point 1: There were previous studies reporting similar data (as mentioned in the
Introduction section). The authors should highlight the novelty of this study and its
contribution to the field.
Response 1: Thanks for the comment. We have rewritten this part in the end of
Introduction section to include the study novelty and the contribution to the field as
follows.
Previous findings indicate that an effective means of inducing sweating to optimize
the amount of excreted heavy metals should be identified, however, studies have
rarely systematically compared the influence of different stimulus of sweating on the
concentration of heavy metal in sweat. The objective of this study was thus to
preliminarily examine the difference in the excretion of heavy metals under two
sweating conditions. This study performed a simulated sweating test on 12 healthy,
young participants (6 men and 6 women) and collected their sweat under two
sweating conditions (dynamic running and a static sauna environment). Sweat
compositions were analyzed for the excretion of different heavy metals (Ni, Pb, Cu,
As, and Hg) under the two conditions. We hypothesized that the excretions of five
heavy metals in sweat under two sweating environments could be distinct based on
different physiological mechanisms. (Line 66‐76)
This preliminarily study suggests that individuals experiencing adverse effects due to
toxic elements or those who have regular exposure to or accretion of toxicants may
consider sweating through dynamic exercise (e.g., running) as an option. (Line 305‐
307)
The results can serve as a reference for sweat detoxification while performing daily
activities. (Line 315‐316)

Point 2: The results should be compared with those in previous studies where relevant.
Response 2: as suggested, we have added descriptions to discuss about the results of
the previous studies as follows.
These variable sources may lead to considerable individual differences in heavy metal
compositions. In addition, the experimental setting also determined the interpretative
data. Tang et al. [11] found the excretions of Cu 2.59 μg/L and Pb 0.63 μg/L in sweat
under arduous exercise, while Genuis et al. [24] obtained those of Cu 652.49 μg/L and
Pb 25.67 μg/L under infrared or steam sauna. The large variance in excretion of each
metal between studies demonstrated that not only the individual differences but also
the systematic experiments influenced the results. (Line 278‐284)

Point 3: Line 64‐65: Analysis methods for each metal should be in detail (lines 152‐
156). Relevant references should be cited.
Response 3: Analysis methods for each metal have been added in the revision and
relevant references were also cited as follows.
All composition analyses were acidified with HNO3 and completed within 32 h of
each trial. The sweat samples were analyzed for Ni, Pb, Cu, As, and Hg levels
complying with the standard methods [20,21]. Samples were measured in duplicate.
Ni, Pb, and Cu concentrations were analyzed using ICP‐AES (PerkinElmer, Avio 200,
IL, USA) with method detection limits (MDLs) of 1.1, 3.0, and 1.5 μg/L, respectively.
Analyses of As and Hg were performed using ICP‐MS (Agilent 7800, CA, USA) with
MDLs of 0.03 and 0.05 μg/L, respectively. Method detection limits (MDLs) for each
metal were conducted in our laboratory following the standard methods [22,23].
MDLs were calculated as three times the standard deviation of a set of method blanks
and the values of quantification limits of the study. The calibrated standard solutions
of Ni, Pb, and Cu were prepared ranging from 0‐500μg/L; that of As and Hg from 0‐
5μg/L, which ranges covered the concentration of metals in sweat founded in
literatures [15,24]. (Line 143‐155, References 20‐23)

Point 4: Please clarify the MDLs (lines 154‐156). Were they from previously reported
data? Were there values of quantification limits?
Response 4: As responded to the point 3, method detection limits (MDLs) for each
metal were conducted in our laboratory following the standard methods [22,23].
MDLs were calculated as three times the standard deviation of a set of method blanks
and the values of quantification limits of the study. The calibrated standard solutions
of Ni, Pb, and Cu were prepared ranging from 0‐500μg/L; that of As and Hg from 0‐
5μg/L, which ranges covered the concentration of metals in sweat founded in
literatures [15,24]. (Line 150‐155)

Point 5: Table 2: please clarify the numbers presented in a table footnote. Were they
Means (SDs)?
Response 5: As suggested, a table footnote has been added. (Table 2)

Point 6: The data in Figure 2 should be presented as boxplots, which clearly show the
range, median, and mean values of each dataset. It will be more informative to readers.
Response 6: Many thanks for the valuable comment. We have added Figure 2 to
present the boxplots for each heavy metal in the revision. Because the measurement
amounts among heavy metals are quite different, two scales are thus adopted in the
figure. In addition, the descriptions for the figure have also been added. (Line 201‐206,
Figure 2)

Point 7: The data for Hg were unreliable. The samples should be preconcentrated
before analysis, by which those concentrations below MDLs could be measured
accurately.
Response 7: Thanks for the suggestion. The concentration of Hg in all samples are
detectable and higher than the method detection limits (MDL) of Hg in the study of
0.05 μg/L using ICP‐MS; therefore, the preconcentrated procedure for the sweat
sample was not used in the analysis. (Table 2 and Figure 2)

Reviewer 3 Report

From a generic perspective this study would need follow this major changes. After this corrections it mus be reconsiderare to be published.

1.- It does not follow an argumental line. You must go from the most generic thing to the study objetive. 

2.- You must incorporate the hypothesis.

3.- You must incorporate the main objectives of this study. 

4.- Please. Avoid mentioning authors as much as possible. In the introduction you must explain the most important aspects about sweat under sweat environments and don't explain what the other authors say. Use good reference to give support your ideas. 

5.- Please. You must introduce a specific section for results. Not results and discussion together. 

6.- You must incorporate a practical point of view of the applications on your results and conclusions and you and a future idea should be offered for further investigation but also from a practical vision. 

The idea is not bad but form my personal point of view it needs restructarete the content and the structure. 

Author Response

Response to Reviewer 3 Comments
The authors wish to sincerely appreciate the reviewers for the opportunity given to
improve our manuscript. Please find below the improvement and explanation made
by the authors based on the reviewer’s comments and recommendations.
※ The marked changes were marked by using colored (red) text in the revised manuscript.
From a generic perspective this study would need follow this major changes. After
this corrections it must be reconsiderare to be published.

Point 1: It does not follow an argumental line. You must go from the most generic
thing to the study objective.
Response 1: Thanks for the comment. As suggested, the manuscript has been
reorganized and rephrased to make the narrative more succinct, particularly
Introduction section. These motivated us to develop the study objective as shown in
the end of the Introduction section. (Line 69‐70)

Point 2: You must incorporate the hypothesis.
Response 2: We hypothesized that the excretions of five heavy metals in sweat under
two sweating environments could be distinct based on different physiological
mechanisms. The hypothesis has been added in the end of Introduction section. (Line
74‐76)

Point 3: You must incorporate the main objectives of this study.
Response 3: As responded to the Point 1, the study objective has been added as
follows.
The objective of this study was thus to preliminarily examine the difference in the
excretion of heavy metals under two sweating conditions. (Line 69‐70)

Point 4: Please. Avoid mentioning authors as much as possible. In the introduction
you must explain the most important aspects about sweat under sweat environments
and donʹt explain what the other authors say. Use good reference to give support your
ideas.
Response 4: Thanks for the expert comment. We have checked this issue and corrected
it throughout the manuscript to avoid mentioning the authors’ names directly useless
necessary in some cases.

Point 5: Please. You must introduce a specific section for results. Not results and
discussion together.
Response 5: In the revision, we have separated the original Results and discussion
into Results section and Discussion section independently, and added Figure 2 in
Results section.

Point 6: You must incorporate a practical point of view of the applications on your
results and conclusions and a future idea should be offered for further investigation
but also from a practical vision.
Response 6: As suggested, we have added the further investigation and field
application in the related paragraphs as follows.
However, only 12 healthy young university students were recruited; this sample size
is relatively small. A more extensive study would allow an increase in experimental
settings (various populations, health statuses, and sweating conditions) for future
research. (Line 307‐310)
This preliminarily study suggests that individuals experiencing adverse effects due to
toxic elements or those who have regular exposure to or accretion of toxicants may
consider sweating through dynamic exercise (e.g., running) as an option. (Line 305‐
307)
The results can serve as a reference for sweat detoxification while performing daily
activities. (Line 315‐316)

Point 7: The idea is not bad but form my personal point of view it needs restructarete
the content and the structure.
Response 7: We thank the reviewer wholeheartedly. The comments which we have
conducted carefully and they really enhance and polish the readability and quality of
the manuscript.

Round 2

Reviewer 1 Report

The authors have made all  the suggested changes asked by this reviewer, and in general the new writing is clear and in good English.  the exception is for lines 229-234 (previously 215-216). A suggested revised version of the sentence in response 11 is:

Heavy metals are most abundant exogenously, but small amounts enter the body through food and water.  Most are barely metabolized and their......sweat.  Some heavy metals, however, such as.....are important for life and metabolism. Others, however, are toxic, interfering with protein function and enzyme activities.  Excessive accumulation of toxic heavy metals con even lead to chronic poisoning or death[26]."

Author Response

Response to Reviewer 1 Comments (R2)

#The marked changes were marked by using colored (red) text in the revision.

Point 1: The authors have made all the suggested changes asked by this reviewer, and in general the new writing is clear and in good English.  the exception is for lines 229-234 (previously 215-216). A suggested revised version of the sentence in response 11 is:

Heavy metals are most abundant exogenously, but small amounts enter the body through food and water.  Most are barely metabolized and their......sweat. Some heavy metals, however, such as.....are important for life and metabolism. Others, however, are toxic, interfering with protein function and enzyme activities.  Excessive accumulation of toxic heavy metals con even lead to chronic poisoning or death [26]."

Response 1: Thanks for the valuable comment. In the revision, we have revised the descriptions as suggested. (Line 229-234)

The authors wish to sincerely appreciate the reviewer again for the valuable comments to improve our manuscript.

Reviewer 2 Report

The manuscript was appropriately revised and can be accepted for publication.

Author Response

The authors wish to sincerely appreciate the reviewer again for the valuable comments to improve our manuscript.

Reviewer 3 Report

From my personal point of view this documents has the formal aspecto to be publishes. It is not a originality and top article but with the changed done I think this document can be considered to be accepted. 

Author Response

(The authors gave the same response as above.)
